# CAUSAL TESTING OF REPRESENTATION SIMILARITY METRICS

## ABSTRACT

Representation similarity metrics are widely used to compare learned representations in neural networks, as is evident in extensive literature investigating metrics that accurately captures information encoded in the network. However, aiming to capture all of the information available in the network may have little to do with what information is actually used by the network. One solution is to experiment with causal measures of function. By ablating groups of units thought to carry information and observing whether those ablations affect network performance, we can focus on an outcome that causally links representations to function.

In this paper, we systematically test representation similarity metrics to evaluate their sensitivity to causal functional changes induced by ablation. We use network performance changes after ablation as way to causally measure the influence of representation on function. These measures of function allow us to test how well similarity metrics capture changes in network performance versus changes to linear decodability. Network performance measures index the information used by the network, while linear decoding methods index available information in the representation.

We show that all of the tested metrics are more sensitive to decodable features than network performance. Within these metrics, Procrustes and CKA outperform regularized CCA-based methods on average. Although Procrustes and CKA outperform on average, for AlexNet, Procrustes and CKA no longer outperform CCA methods when looking at network performance. We provide causal tests of the utility of different representational similarity metrics. Our results suggest that interpretability methods will be more effective if they are based on representational similarity metrics that have been evaluated using causal tests.

## 1 INTRODUCTION

Neural networks already play a critical role in systems where understanding and interpretation are paramount like in self-driving cars and the criminal justice system. To understand and interpret neural networks, representation similarity metrics have been used to compare learned representations between and across networks (Kornblith et al. (2019); Raghu et al. (2017); Morcos et al. (2018b); Wang et al. (2018); Li et al. (2015); Feng et al. (2020); Nguyen et al. (2020)). Using these similarity metrics, researchers evaluate whether networks trained from different random initializations learn the same information, whether different layers learn redundant or complementary information, and how different training data affect learning (Kornblith et al. (2019); Li et al. (2015); Wang et al. (2018)). Apart from helping to answer these fundamental questions, similarity metrics have the potential to provide a general-purpose metric over representations Boix-Adsera et al. (2022).

What it means for two representations to be similar, however, is not straightforward. Many similarity metrics have been proposed with different underlying assumptions and strategies for comparing representation spaces. For example, some similarity metrics are invariant under linear transformations while others are not (see Kornblith et al. (2019) for a theoretical comparison). These different assumptions and strategies can lead to quantitatively different predictions. For instance, Ding et al. (2021) show that certain metrics are insensitive to changes to the decodable information present in representations. In another study, Davari et al. (2022) demonstrate that the centered kernel alignment metric predicts a high similarity between random and fully trained representations. It is therefore

unclear which representation similarity metrics capture the most important information from representations and further tests are needed to evaluate them.

What important pieces of information do similar representations share? Previous studies into similarity metrics have assumed that similar representations share linearly decodable information Boix-Adsera et al. (2022); Ding et al. (2021); Feng et al. (2020). To measure the linearly decodable information in a representation, researchers usually train linear probes for downstream tasks on learned representations and compare the results. However, the features of a representation that carry the most information may not be those actually used by the network during inference. Studies that remove features from representations in trained networks have revealed a weak link between the relevance of a feature for decoding and its effect when removed from the network Meyes et al. (2020); Zhou et al. (2018); Donnelly & Roegiest (2019); Morcos et al. (2018b). Hayne et al. (2022) recently showed that linear decoders specifically cannot single out the features of representations actually used by the network. Consequently, two representations that are equally decodable using linear probes may not actually be equal from the point of view of network performance. This distinction is crucial for neural network interpretability where the aim is to develop human-understandable descriptions of how neural networks actually rely on their internal representations.

For the purpose of interpreting neural network function, we suggest that representations should be judged as similar if they cause similar effects in a trained network. To observe these causal effects, previous studies have removed features from a representation, a process called ablation, and observed the effects LeCun et al. (1989). In this paper, we use ablation to evaluate how closely representation similarity metrics are related to causal function. We first ablate groups of units from AlexNet and MobileNet, compare the original representations to the ablated representations using representation similarity metrics, and then compare metric outputs to the changes seen for linear probe decoding or network performance. This way we can test how well representational changes from ablations are captured by representational similarity metrics by comparing those metrics to changes in linear decoding and causal differences in network performance.

Linear probes measure how much task-specific information is directly decodable from a given representation. On the other hand, network performance measures quantify how the network trained on the same task uses a given representation. Finally, we test how well representation similarity metrics capture these two changes. By directly comparing linear probe accuracies and network performances on the same task we can answer questions like: how much more sensitive are representation similarity metrics to the non-causal linear properties of representations compared to the causal non-linear properties used by the network? Answering these questions may help in the development of interpretability methods that are increasingly sensitive to actual network function.

In this work, we show that CKA, Procrustes, and regularized CCA-based representation similarity metrics predict causal network performance changes significantly worse than non-causal decoding changes. We also show that, on average, Procrustes and CKA outperform regularized CCA-based methods. However, Procrustes and CKA do not outperform regularized CCA-based metrics on all network and functional measure combinations. Overall, our results suggest that interpretability methods will be more effective if they are based on representational similarity metrics that have been evaluated using causal tests. In general, this paper documents the following contributions:

- We introduce a causal test of the utility of representation similarity metrics. We find that five popular representation similarity metrics are significantly less sensitive to network performance changes induced by ablation than linearly decodable changes.

- Within the tested metrics, we show that Procrustes and CKA tend to outperform regularized CCA-based methods, but that tests using linear probes and network performance based functional measures can produce different results in different networks.

## 2 METHODS

In Section 2.1, we describe the statistical testing methodology used in our experiments. In Section 2.2, we introduce the representation similarity measures we evaluate and reformulate them for use on high dimensional representations. In Section 2.3, we describe how we use ablation to produce representations with different functional properties. Finally, in Section 2.4, we describe how we

use linear probe decoding deficits and class-specific performance deficits to measure decodable and downstream network changes, respectively, in the ablated representations.

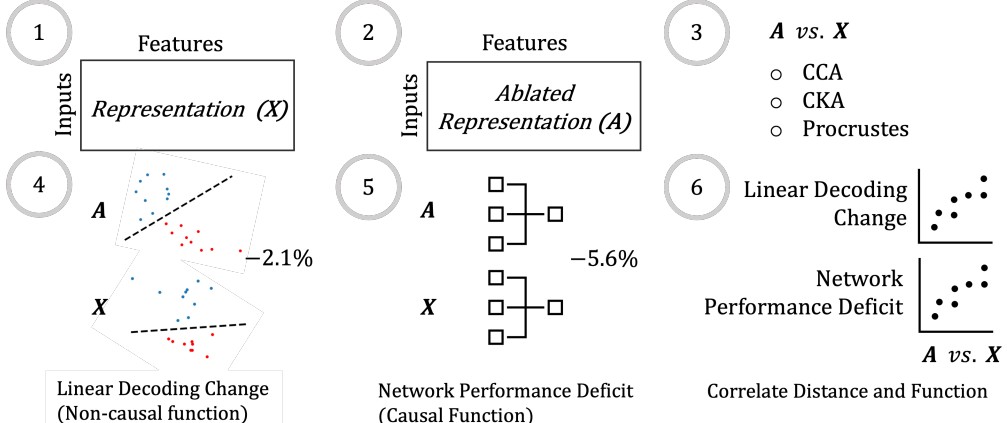

Figure 1: **Elements of the experimental design. 1)** A representation matrix $\boldsymbol{X}$ is generated from a layer in a trained network. **2)** Ablations are applied by deleting features from the representation to generate the ablated matrix $\boldsymbol{A}$. Representation matrices $\boldsymbol{X}$ and $\boldsymbol{A}$ are compared in three ways. **3)** A representation metric similarity is calculated between them. **4)** Linear probes are fit to each to decode a target class and the linear probe accuracies are compared. **5)** The representations are fed back into the network and the network performance difference between them is calculated. **6)** By comparing metric similarities with both linear probe decoding changes and network performance changes across many ablations (represented with multiple points in the correlation plots in panel 6), we can assess to what extent each metric captures non-causal and causal measures of function.

## 2.1 STATISTICAL TESTING

Assume $\boldsymbol{A} \in \mathbb{R}^{n \times p_1}$ represents a matrix of activations for $p_1$ neurons given $n$ examples, and $\boldsymbol{B} \in \mathbb{R}^{n \times p_2}$ represents a matrix of activations for $p_2$ neurons given the same $n$ examples. The matrices $\boldsymbol{A}$ and $\boldsymbol{B}$ are called representation matrices and have been preprocessed to have centered columns. Let $d(\boldsymbol{A}, \boldsymbol{B})$ denote a representation similarity metric that returns zero if and only if $\boldsymbol{A} = \boldsymbol{B}$ and for which $d(\boldsymbol{A}, \boldsymbol{B}) = d(\boldsymbol{B}, \boldsymbol{A})$. These metrics do not satisfy the triangle inequality and are therefore not formal distance metrics. For simplicity, we will refer to them as metrics in this work.

To quantify functional differences between representation matrices, we use functional behavior measures. Formally, let $f : \mathbb{R}^{n \times p} \rightarrow \mathbb{R}$ denote a functional behavior measure that, given a representation matrix, returns a scalar measure of the representation's role in function. In this study, we utilize two functionality measures, class-specific linear decoding accuracy ($f_{\text{decoding}}$) and class-specific network performance ($f_{\text{performance}}$). In the case of linear decoding accuracy, $f_{\text{decoding}}$ returns the average linear probe accuracy achieved from decoding a target class identity from a representation matrix. On the other hand, $f_{\text{performance}}$ returns the average classification performance for a target class achieved by feeding the representation matrix to the network at the appropriate layer. More details are presented in Section 2.4.

As in Ding et al. (2021), we aim to statistically test representation similarity metrics using the same methodology according to their paper:

1. Gather $S$, a set of representation matrices that vary along one or more dimensions. We note that $S$ contains unablated as well as ablated representation matrices (see Section 2.3).

2. Select an unablated reference matrix $\boldsymbol{X} \in S$.

3. Compute the following for all $\boldsymbol{A}_i \in S$ where $\boldsymbol{A}_i \neq \boldsymbol{X}$ :
    - D $= d(\boldsymbol{X}, \boldsymbol{A}_i)$
    - F $= |f(\boldsymbol{X}) - f(\boldsymbol{A})|$

4. Compute the correlation between D and F using Spearman's correlation.

This procedure quantifies the extent to which the representation similarity metrics, $d(\cdot)$, capture the functionality differences, as measured by $f(\cdot)$, produced by ablating the representation matrix. A high Spearman's correlation value between a metric's computed representation similarities and the functionality differences produced by ablation implies that the chosen metric is *sensitive* to the chosen functionality. Whereas, a low correlation implies the opposite: that the chosen metric is not sensitive to the chosen functionality.

## 2.2 REPRESENTATION SIMILARITY METRICS

As in Ding et al. (2021), we study three main representation similarity metrics: centered kernel alignment (CKA), Procrustes, and projection-weighted canonical correlation analysis (PWCCA).

**Centered kernel alignment (CKA)** is based on the idea that similar representations also have similar relations between examples. In other words, representation matrices that store images of lettuce and rabbits using similar vectors should be more similar to each other than with representation matrices that encode images of lettuce and dinner plates using similar vectors. This idea leads Kornblith et al. (2019) to formulate linear CKA, which uses a linear kernel to compare example vectors (henceforth referred to as just CKA):

$$d_{\text{CKA}}(\boldsymbol{A}, \boldsymbol{B}) = 1 - \frac{\|\boldsymbol{A}^{\text{T}}\boldsymbol{B}\|_{\text{F}}^2}{\|\boldsymbol{A}^{\text{T}}\boldsymbol{A}\|_{\text{F}}\|\boldsymbol{B}^{\text{T}}\boldsymbol{B}\|_{\text{F}}} \tag{1}$$

where $\|\cdot\|_{\text{F}}$ is the Frobenius norm and $\|\boldsymbol{A}^{\text{T}}\boldsymbol{B}\|_{\text{F}}^2$ derives from the following relation:

$$\langle \text{vec}(\boldsymbol{A}\boldsymbol{A}^{\text{T}}), \text{vec}(\boldsymbol{B}\boldsymbol{B}^{\text{T}}) \rangle = \text{tr}(\boldsymbol{A}\boldsymbol{A}^{\text{T}}\boldsymbol{B}\boldsymbol{B}^{\text{T}}) = \|\boldsymbol{A}^{\text{T}}\boldsymbol{B}\|_{\text{F}}^2 \tag{2}$$

Relation 2 shows that the similarity between pairwise example similarity matrices (far left) is equal to the squared Frobenius norm of the feature covariance matrix between representations (far right). Kornblith et al. (2019) use this relation to form Equation 1 which measures the normalized similarity between the example similarity matrices of $\boldsymbol{A}$ and $\boldsymbol{B}$. Unfortunately, computing and storing either $\boldsymbol{A}^{\text{T}}\boldsymbol{B}$, $\boldsymbol{A}^{\text{T}}\boldsymbol{A}$, or $\boldsymbol{B}^{\text{T}}\boldsymbol{B}$ can be prohibitively expensive when both $p_1$ and $p_2$ grow too large. Therefore we reformulate Equation 1 using relation 2 and the fact that $\|\boldsymbol{X}^{\text{T}}\boldsymbol{X}\|_{\text{F}} = \|\boldsymbol{X}\boldsymbol{X}^{\text{T}}\|_{\text{F}}$ into:

$$d_{\text{CKA}}(\boldsymbol{A}, \boldsymbol{B}) = 1 - \frac{trace(\boldsymbol{A}\boldsymbol{A}^{\text{T}}\boldsymbol{B}\boldsymbol{B}^{\text{T}})}{\|\boldsymbol{A}\boldsymbol{A}^{\text{T}}\|_{\text{F}}\|\boldsymbol{B}\boldsymbol{B}^{\text{T}}\|_{\text{F}}} \tag{3}$$

This reformulation allows us to use CKA on layers with at least 100 times more neurons than previous studies.

**Procrustes** is an analytical solution to the orthogonal Procrustes problem which involves finding a right rotation of matrix $\boldsymbol{B}$ that is as close as possible to $\boldsymbol{A}$ as measured by the Frobenius norm:

$$d_{\text{Procrustes}}(\boldsymbol{A}, \boldsymbol{B}) = \|\boldsymbol{A}\|_{\text{F}}^2 + \|\boldsymbol{B}\|_{\text{F}}^2 - 2\|\boldsymbol{A}^{\text{T}}\boldsymbol{B}\|_* \tag{4}$$

where $\|\cdot\|_*$ is the nuclear norm. As with CKA, $\boldsymbol{A}^{\text{T}}\boldsymbol{B}$ needs to be replaced to lighten the computational cost of working with large layers. Therefore, we utilize the fact that the nuclear norm of a matrix is the sum of its singular values to reformulate Procrustes:

$$d_{\text{Procrustes}}(\boldsymbol{A}, \boldsymbol{B}) = \|\boldsymbol{A}\|_{\text{F}}^2 + \|\boldsymbol{B}\|_{\text{F}}^2 - 2\sum_{i}^{n} \sqrt{\lambda_i(\boldsymbol{A}\boldsymbol{A}^{\text{T}}\boldsymbol{B}\boldsymbol{B}^{\text{T}})} \tag{5}$$

where $\lambda_i(\boldsymbol{X})$ represents the $i^{\text{th}}$ eigenvalue of matrix $\boldsymbol{X}$. Again, this reformulation saves us from manipulating the large $p_1 \times p_2$ matrix by replacing it with a much smaller $n \times n$ matrix (assuming $n \ll p_1, p_2$).

**Projection-weighted canonical correlation analysis (PWCCA)** is a special case of canonical correlation analysis (CCA) proposed by Morcos et al. (2018a). CCA itself provides a solution to the problem of linearly projecting $\boldsymbol{A}$ and $\boldsymbol{B}$ into a shared subspace where their correlations are maximized. CCA finds $\min(p_1, p_2)$ pairs of weight vectors $(\boldsymbol{w_A}, \boldsymbol{w_B})$ and the resulting correlation

induced by projecting $\boldsymbol{A}$ and $\boldsymbol{B}$ using the $i^{\text{th}}$ weight vector is:

$$\rho_i = \max_{\boldsymbol{w}_{\boldsymbol{A}}^i, \boldsymbol{w}_{\boldsymbol{B}}^i} \text{corr}(\boldsymbol{A}\boldsymbol{w}_{\boldsymbol{A}}^i, \boldsymbol{B}\boldsymbol{w}_{\boldsymbol{B}}^i)$$

$$\text{subject to} \quad \forall_{j<i} \quad \boldsymbol{A}\boldsymbol{w}_{\boldsymbol{A}}^i \perp \boldsymbol{A}\boldsymbol{w}_{\boldsymbol{A}}^j \tag{6}$$

$$\boldsymbol{B}\boldsymbol{w}_{\boldsymbol{B}}^i \perp \boldsymbol{B}\boldsymbol{w}_{\boldsymbol{B}}^j$$

where the $\rho_i$ is maximized subject to the constraint that the subspace features be orthogonal. Equation 6 can be solved for by performing singular value decomposition on $(\boldsymbol{A}^{\text{T}}\boldsymbol{A})^{-1/2}\boldsymbol{A}^{\text{T}}\boldsymbol{B}(\boldsymbol{B}^{\text{T}}\boldsymbol{B})^{-1/2}$ where the singular values are equal to the correlations ($\rho_i \ \forall \ i \in [1,...,\min(p_1, p_2)]$).

However, the inverses of the feature covariance matrices do not exist when the number of neurons exceeds the number of examples. For these cases, we can use regularized or "ridge" CCA (Vinod (1976)), which applies an $L_2$ penalty to the weight vectors and can be solved by performing SVD on $(\boldsymbol{A}^{\text{T}}\boldsymbol{A} + \kappa_{\boldsymbol{A}}\boldsymbol{I})^{-1/2}\boldsymbol{A}^{\text{T}}\boldsymbol{B}(\boldsymbol{B}^{\text{T}}\boldsymbol{B} + \kappa_{\boldsymbol{B}}\boldsymbol{I})^{-1/2}$. However, we again run into the problem that the feature covariance matrices are too costly to compute for large layers. So, we employ the "kernel trick" introduced by Kuss & Graepel (2003) and Hardoon et al. (2004) and refined by Tuzhilina et al. (2021) which allows us to substitute into the above expression $\boldsymbol{R_A}$ for $\boldsymbol{A}$ where $\boldsymbol{R_A}$ represents an $n \times n$ matrix recovered by applying SVD on $\boldsymbol{A}$, i.e. $\boldsymbol{A} = \boldsymbol{R_A}\boldsymbol{V}_{\boldsymbol{A}}^{\text{T}}$. The same trick can be applied to $\boldsymbol{B}$. These substitutions allows us to work with much smaller matrices and make CCA computationally tractable for large layers. The only drawback is that if we apply the "kernel trick" to both matrices, we recover only $n$ canonical correlations rather than $\min(p_1, p_2)$.

Mean CCA (as used by Raghu et al. (2017)) and mean squared CCA (Ramsay et al. (1984)) average raw and squared correlations recovered through CCA, respectively. However, PWCCA re-weights each correlation value by its importance for the underlying representation. Formally, if representation matrix $\boldsymbol{A}$ has neuron activation vectors $[\boldsymbol{z}_1, ..., \boldsymbol{z}_{p_1}]$ and CCA vectors $[\boldsymbol{h}_1, ..., \boldsymbol{h}_n]$, then PWCCA computes a weighted mean as:

$$d_{\text{PWCCA}}(\boldsymbol{A}, \boldsymbol{B}) = 1 - \sum_{i=1}^{n} \tilde{\alpha}_i \rho_i \quad \text{s.t.} \quad \alpha_i = \sum_j |\langle \boldsymbol{h}_i, \boldsymbol{z}_j \rangle|$$

where $\tilde{\alpha}_i$ is a normalized version of $\alpha_i$. The preceding CCA reformulations represent regularized forms of CCA-based metrics useful for high-dimensional representations, which we refer to as regularized CCA-based metrics in the text, but just CCA metrics in the figures for sake of brevity.

## 2.3 ABLATION

To study representation functionality changes, we needed a method for manipulating representations to reliably produce network performance deficits at the output of the network. To reliably produce network performance deficits, we followed the procedure of Hayne et al. (2022). Specifically, for each of the 10 or 50 randomly chosen target classes and layer of the CNNs we tested, AlexNet and MobileNetV2, we first projected every neuron onto two dimensions: class selectivity and activation magnitude. Then, we constructed a grid to overlay on the activation space so that each cell of the grid contained the same number of neurons. To supply the set $S$ of representation matrices from Section 2.1, we ablated one cell of neurons at a time by setting the activation values for those neurons to zero.

## 2.4 FUNCTIONALITY MEASURES

After collecting the set $S$ of ablated representation matrices, we sought to compare two functionality measures. First, we fit linear probe decoders to each representation matrix in $S$. With the linear probes we aimed to decode the identity of one target class, so we fit simple logistic regression classifiers to distinguish the target class representations from all other representations in the representation matrix. Because many of the CNN layers contained thousands of features, we randomly selected 100 neurons as features in the logistic regression model and averaged the training accuracy over 200 repetitions as in Alain & Bengio (2016). Intuitively, the accuracies measure how much

information a representation matrix contains about the target class. We refer to the changes induced in the decodability of the target class by ablation as decoding accuracy deficits.

In contrast to linear decoding accuracies, we also measured the network's class-specific classification performance when utilizing the representations in the representation matrix during inference. Specifically, we recorded the average classification rank of the target class. For instance, if the CNN's softmax layer predicted that the Junco bird class was the third most likely class given an image of a Junco bird, then that image received a rank of three. The average class-specific ranks intuitively measure how functionally useful the representations from the representation matrix were to the network. We refer to the changes induced in class-specific classification rank by ablation as network performance deficits.

## 3  RESULTS

In this section, we detail the results of aggregating our statistical tests across each layer and class from AlexNet and MobileNetV2. We conducted a hierarchical linear model to evaluate the prediction of the rank correlation values as a function of two factors: 1) *metric*, i.e. CKA, Procrustes, and regularized CCA-based methods and 2) *functionality*, i.e. network performance deficits and decoding accuracy deficits. To allow for comparable correlation coefficients, we fisher-z transformed each rank correlation value, then, averaged them across layers for each each metric and functionality. Afterwards, we inverse transformed the fisher-z back to correlation coefficients. We modeled classes as random units. To sum, the transformed correlation values were modeled as a function of metric type and functionality measure. All analyses were conducted using the "lme4" package in R (Bates et al. (2014)). Figures were generated using "ggplot2" (Wickham (2016), "raincloudplots" (Allen et al. (2021)), and "smplot" package (Min & Zhou (2021)) in R.

### 3.1  REPRESENTATION SIMILARITY METRICS ARE SIGNIFICANTLY LESS SENSITIVE TO CAUSAL FUNCTION

Figure 2a shows the distribution of correlation values across functionality measures. In both cases, the rank correlation values of both functionality measures are significantly different from 0, suggesting that the correlation values are sensitive to functional behavioral changes in the representations, ($F(1,49) = 470.37$, $p < .001$ for AlexNet, $F(1,9) = 1464.06$, $p < .001$ for MobileNetV2). The analysis of interest demonstrates these correlation values are significantly different between network performance deficits and decoding accuracy deficits, when averaging across the five similarity metrics — CKA, Procrustes, and regularized CCA-based metrics ($F(1,441) = 633.85$, $p < .001$). In other words, there is a significant main effect of functionality. Likewise, Figure 2b shows the same results for MobileNetV2 ($F(1,81) = 5770.04$, $p < .001$).

### 3.2  CKA AND PROCRUSTES TEND TO OUTPERFORM OTHER METRICS

Figures 3a and 3b show that CKA and Procrustes have, on average, higher rank correlation values compared to the other CCA-based metrics ($t(441) = 6.33$, $p < .001$ for AlexNet; $t(81) = 15.70$, $p < .001$ for MobileNetV2). Again, higher rank correlation values indicate a higher coupling between functionality measures and representation similarity metrics, thereby suggesting that CKA and Procrustes are better metrics at capturing functionality.

### 3.3  FUNCTIONALITIES PRODUCE DIFFERENT RESULTS FOR DIFFERENT NETWORKS

Figure 4 (top) shows a significant interaction between functionality measures and similarity metrics for AlexNet ($F(4,441) = 12.21$, $p < .001$). Figure 4 (bottom) shows the same model for MobileNetV2, however, interaction is non-significant ($F(4,81) = .88$, $p = .48$) In the case of AlexNet, with the plotted means, it is evident that the interaction is driven by the metric differences within decoding accuracy deficits. As for the similarity metrics within network performance deficits, no single metric outperforms the other; instead, each metric achieves a similarly low correlation amongst the network performance deficit group. As for the decoding accuracy deficit group, on the other hand, CKA and Procrustes outperform the CCA-based metrics ($t(441) = 9.31$, $p < .0001$). In the case of MobileNetV2, there is no significant interaction. CKA and Procrustes outperform the CCA-based

metrics on both decoding ($t(81) = 12.41$, $p < .0001$) and network performance tests ($t(81) = 9.79$, $p < .0001$), thereby canceling out an interaction effect.

## 4    DISCUSSION

In this work, we systematically test representation similarity metrics on ImageNet-trained CNNs to determine how sensitive they are to network performance and decoding changes in representations. To facilitate these tests, we reformulate previously proposed similarity metrics to make them computationally tractable for use on high-dimensional representations. Our tests revealed that, while similarity metrics are significantly sensitive to network performance and decoding measures of functionality, they are significantly less sensitive to network performance. Additionally, in most of our tests CKA and Procrustes significantly outperform CCA-based methods. However, we do find that different networks and different functionality choices can eliminate their advantages.

Previously, Ding et al. (2021) performed some general stress tests on representation similarity metrics. In their work they showed that PWCCA distances were easily influenced by changes to random seed, predicting that the same layers in two different networks were more similar to other distant layers in the network than to each other. In another test, the CKA metric suffered from a different problem where it failed to distinguish representations from their low rank counterparts. Davari et al. (2022) pointed out other weaknesses associated with CKA. Among other findings, they demonstrated that CKA judged random and fully trained representations to be highly similar, especially in early layers of a network. Both studies used functional and intuitive tests for evaluation.

Although our work utilizes a framework introduced by Ding et al. (2021), we aim to test a fundamentally different hypothesis not addressed by their tests. Specifically, we aim to test how sensitive representation similarity metrics are to the functional properties of a representation actually used by a trained network during inference. In our tests we directly compare linear probe accuracy changes on in-distribution inputs to network performance changes on the same inputs after ablation. On the other hand Ding et al. (2021) do not test this direct comparison. Instead, they perform more general stress tests on similarity metrics using either in-distribution linear probes or out-of-distribution net-

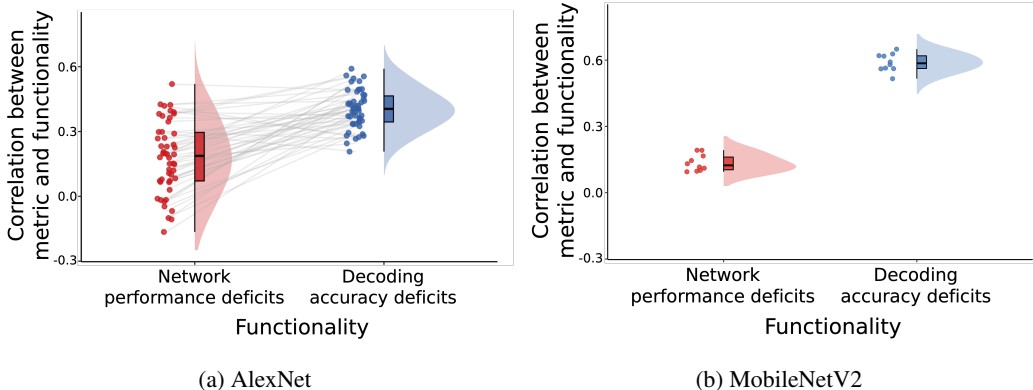

(a) AlexNet                                                       (b) MobileNetV2

Figure 2: **Correlation values overall are significantly sensitive to ablation changes in the representations, however, decoding accuracy has a higher value than network performance deficits.** This figure shows the distribution of rank correlation values between functionality measures, i.e., the main effect of functionality measure on AlexNet (a) and MobileNetV2 (b). Each data point represents a value from the ten classes; within each class, we average the five metrics of CKA, Procrustes, PWCCA, mean CCA and mean squared CCA. The distribution is further depicted by the violin plots and boxplots, which illustrates the median and upper/lower quartile of the distribution. Overall, both functionality measures are significantly different from 0, indicating that the rank correlation values are sensitive to changes produced by ablations in the representations. Further examining the difference between functionality measures, we see that the rank correlation values of decoding accuracy deficit are higher than that of network performance deficits. Note that all CCA-based metrics are regularized in our formulations.

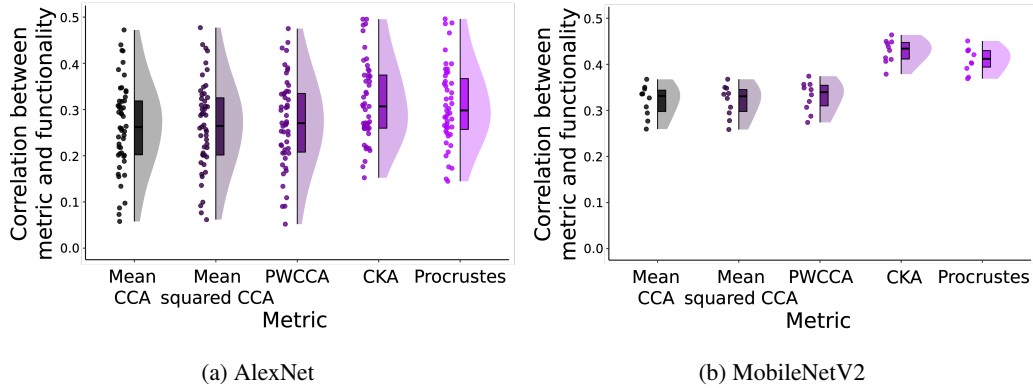

(a) AlexNet  (b) MobileNetV2

Figure 3: **CKA and Procrustes outperform CCA-based metrics.** This figure shows the distribution of rank correlation values across each metric, averaged between functionality measures on AlexNet (a) and MobileNetV2 (b). Within each distribution, each data point represents the average rank correlation value from the ten classes, collapsed across network performance deficits and decoding accuracy deficits. Higher rank correlation values in the CKA and Procrustes condition indicate that these metrics are sensitive to the perturbations in the representations compared to other CCA-based metrics. Note that all CCA-based metrics are regularized in our formulations.

work performance scores and different methods of perturbing representations. Through this direct comparison, we are able to show that similarity metrics are significantly less sensitive to the features of representations that are actually used by the network compared to its decodable features.

All of the representation similarity metrics we tested are significantly less sensitive to causal changes induced by ablation than non-causal decoding changes. This finding reflects the considerations made in developing these similarity metrics. Representation similarity metrics were designed to compare the linear geometric properties of two representation spaces. So, it is not surprising that similarity metrics correlate with changes in decoding accuracies. On the other hand, network performance measures of function reflect how the network utilizes representations. In this case, functionally similar representations are those representations that remain similar after a series of non-linear transformations through layers of the network. Perhaps it is not surprising that this non-linear notion of similarity is harder to capture using current similarity metrics. However, it is the ultimate goal of interpretability to link representation and non-linear network function.

Like previous studies, we show that some metrics tend to outperform others. Ding et al. (2021) previously showed that Procrustes tends to outperform CKA and CCA-based methods on language models. In our experiments on larger image classification models, both CKA and Procrustes tend to perform better than regularized CCA-based methods adapted for larger representations. These results are hinted at by Ding et al. (2021) who show that CKA and Procrustes perform well on image decoding tests, but omit comparisons to CCA-based metrics. Interestingly, CKA and Procrustes did not outperform other methods in all of our experiments. For AlexNet, network performance tests show that all the similarity metrics equally capture network function (Figure 4). In other words, the best representation similarity metric can change according to the network tested and the functionality used. To employ representation similarity metrics for interpretability, metrics should be developed that can capture the functional properties of representations across many networks. These causal tests can help future interpretability studies identify similarity metrics that are sensitive to causal function.

## 5 LIMITATIONS

We acknowledge the following limitation in this work. Reformulating CCA-based measures to accommodate representations with more neurons than examples required using a regularized version of CCA called "ridge" CCA. In utilizing "ridge" CCA we had to choose regularization penalties to apply to each representation matrix. These penalties were chosen to be consistent across all tests, but

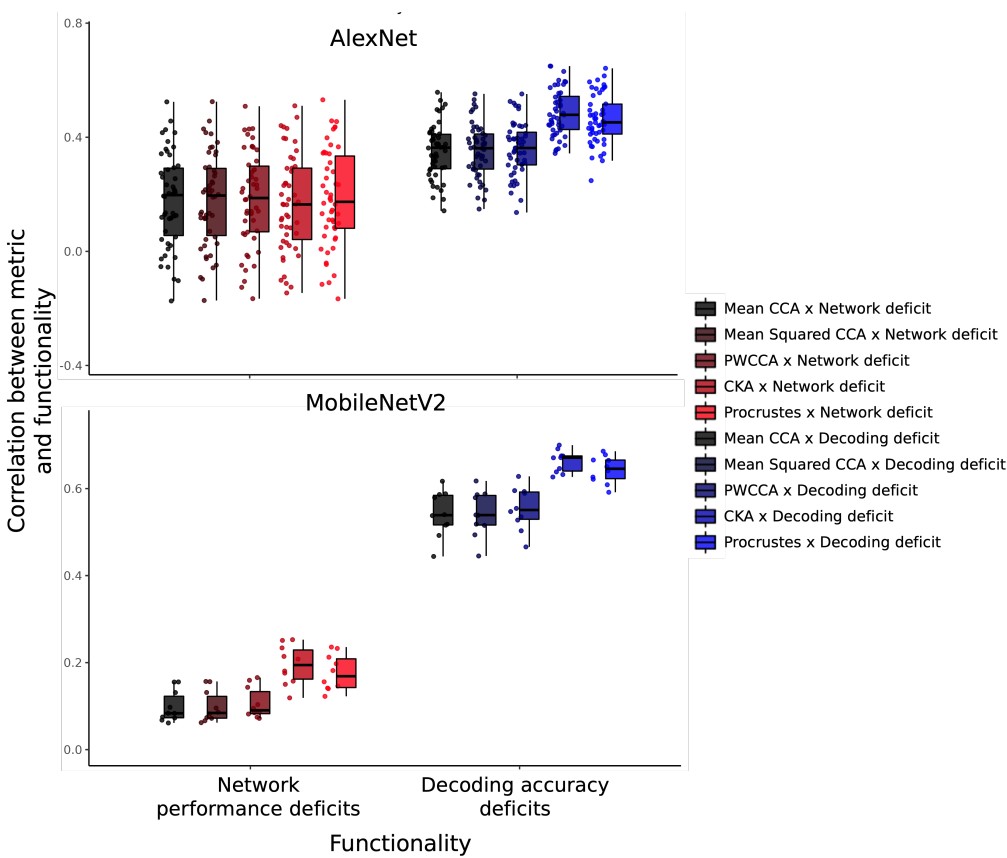

Figure 4: **Pattern of similarity metrics differs depending on functionality measure.** This figure shows a significant interaction between functionality and metric for Alexnet (top) and a non-significant interaction for MobileNetV2 (bottom). In the case of AlexNet, each of the five metrics behave differently depending on functionality type: the five similarity metrics within the network performance deficits condition are not significantly different, whereas metrics do differ within the decoding accuracy deficits condition. Specifically, the CKA and Procrustes metric is significantly different from the CCA-based metrics, only in the decoding accuracy deficit condition. In contrast, in the case of MobileNetV2, this interaction is no longer prominent: CKA and Procrustes consistently outperform CCA-based measures on both functionality tests in MobileNetV2. Note that all CCA-based metrics are regularized in our formulations.

were not cross-validated. Future works would benefit from testing more penalty settings to explore their effect on similarity results.

# 6 CONCLUSION

Taken altogether, the results of this study suggest that representation similarity metrics may already serve well as tools for comparing the geometries of representational spaces, but could be enhanced in order to capture non-linear network function. Similarity metrics achieve high correlations with linear probe decoding accuracy changes in a representation induced by ablation. This sensitivity reveals that existing similarity metrics do a good job of predicting how useful two representations will be for linear downstream tasks. However, trained networks do not necessarily use the features of a representation that are relevant for linear decoding during inference Hayne et al. (2022); Zhou et al. (2018); Meyes et al. (2020); Donnelly & Roegiest (2019). This discrepancy reveals that representation similarity metrics could be improved by taking into account the non-linear features of representations used during inference.

## 7 REPRODUCIBILITY STATEMENT

To ensure the reproducibility of the current work, we have included the code used to conduct and run each of the experiments with the submission. In addition to the code, we have also included in the submission the raw data collected during each of our conducted experiments. At the request of reviewers or other interested parties, additional data sources (i.e. the exact image set used, the representation matrices, etc.) can be made available.

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

# A    APPENDIX

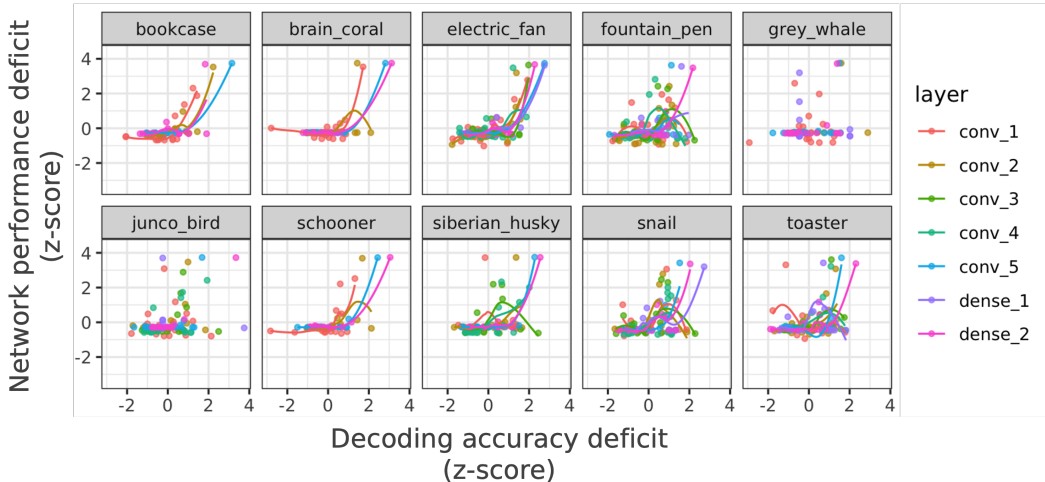

Figure 5: **Decoding accuracy and network performance deficits correlate in later layers of AlexNet.** This figure shows the the relationship between network performance deficit and decoding accuracy deficit across layers and classes in AlexNet, using Local Polynomial Regression Fitting. In later layers of the network, ablations that produce high decoding accuracy deficits also tend to produce high network performance deficits (as illustrated by the fitted lines in the later layers). This relationship does not typically hold for early layers. Note that some points are missing in the case where ablations for certain layer and class combinations did not produce decoding deficits.

