# OpenReview forum: "Grounding High Dimensional Representation Similarity by Comparing Decodability and Network Performance"
_ICLR.cc/2023/Conference — Submitted to ICLR 2023_

### Official Review · Reviewer_u5hR · 2022-10-24

**Confidence:** 2
**Correctness:** 2
**Technical Novelty And Significance:** 2
**Empirical Novelty And Significance:** 2
**Recommendation:** 3

**Clarity, Quality, Novelty And Reproducibility:**

### Clarity and Quality
- This paper is not easy to follow. Overall flow of this paper seems to follow [Ding et al. 2021]. But the paper is not typical organization. For example, Conclusion looks far from highlighting the core arguments of the paper.
- There are too many contribution summary. Indeed, some of them is hard to be considered as contributions (4th and 5th)

### Novelty
- Idea on comparing linear decoding and classification accuracy is interesting. But overall flow is based on the previous work.

### Reproducibility
- The authors submitted the source code. However, it is not easy to understand how to conduct the experiments.

**Strength And Weaknesses:**

### Strength
- In-depth analysis on representation similarity is an important and fundmental topic on representation learning. It can contribute to model interpretability.
- Representation similarity metrics for high-dimensional features is also a significant topic.
- Experiments and analysis on ImageNet-scale data are valuable.

### Weakness
- [Major] This paper seems to be basically based on [Ding et al. 2021] and its extension to high-dimensionality by reformulating metrics and linear decoding analysis. However, the advantage of the high-dimensionality is not effectively showed. The relevant information on high-dimensionality is that the authors validate their method on AlexNet and MobileNetV2 trained with ImagNet only. It is hard to find the specific details on high-dimensionality despite the emphasis in the title.
- [Major]  I am not sure high-dimensionality is mandatory in represenation similarity metric analysis because a feature dimension of a single layer of many Transformer-based models is from 768 to 2048. To show the efficacy, experiments on higher dimensional feature similarity might be required. For example, to emphasize the effects of high-dimensionality, the aughors can concate two layers features over several tens of dimensionality.
- [Major]  The authors presented 10 classes results in Figures 2 and 3. It is not clear how to select 10 classes from 1k classes of ImageNet. Overall, the experiemental setup is not clear for reproducibility and understanding even if they provided the source code.
- [Major]  Following [Ding et al. 2021], experiments on multiple seeds are required.
- [Minor] Although the authors argue they validate their method on popular CNN models, AlexNet and MobileNetV2 are not popular nowadays. Because [Ding et al. 2021] presents the results on BERT, the authors need to consider the similar experiments. The experiments on ViT are ok.

**Summary Of The Paper:**

This paper address a representation similarity metric analysis for high-dimension features by reformulating the representation simiarity metrics such as CKA, Procrustes, and CCA-based methods. Also the authors validate the changes by ablated representation by comparing linear decoding and non-linear classification performances. They present the analysis results on two CNNs such as AlexNet and MobileNetV2 traind on ImageNet.


**Summary Of The Review:**

Despite the topic importance, this paper has some room to be improved (See the weakness). However, I am not an expert in this topic, so I mainy decided my score based on paper organization.

---

> ### Author Response · Authors · 2022-11-19
> **Individual reviewer response to u5hR**
>
> Thank you for your positive feedback recognizing the importance of in-depth analysis of representation similarity metrics, similarity analysis on high-dimensional features, and experiments on ImageNet-scale data. We would like to take the opportunity to address your specific concerns directly below:
>
> *“This paper seems to be basically based on [Ding et al. 2021]...”*
>
> We agree that the novelty of our work was not adequately presented in our first manuscript. In our revised manuscript we have directly emphasized our novel contributions. We aim to portray this work as a unique set of experiments that does not extend the work of [1], but instead tests a fundamentally different hypothesis not addressed by the tests in [1]. Specifically, we aim to test how sensitive representation similarity metrics are to network function. Studies that remove features from representations in trained networks have revealed a weak link between the relevance of a feature for decoding and its effect when removed from the network [2-5]. These studies test the causal relevance of information in representations for function.
>
> So, we conducted experiments to determine if representation similarity metrics were sensitive to this causal information. In our tests we directly compare non-causal linear probe accuracy changes on in-distribution inputs to causal network performance changes on the same inputs. On the other hand, [1] do not test this direct comparison. Instead, they perform more general stress tests on similarity metrics using either in-distribution linear probes or out-of-distribution network performance scores. Through this direct comparison, we are able to show that similarity metrics are significantly less sensitive to the causal features of representations that are actually used by the network compared to its decodable features.
>
> *“The advantage of the high-dimensionality is not effectively showed.”*
>
> Because of the novelty of the reformulated metrics necessary for performing our experiments on larger images, we overemphasized the reformulations. As described in our general response, our experiments were structured to test whether existing similarity metrics actually capture functional changes induced by ablation. As outlined in the response to Shared Concern #1, we have emphasized in the revised manuscript the insensitivity of metrics to function and its implications for interpretability as our major contribution. As a result of this restructuring, we instead frame the reformulated metrics as a tool for facilitating our analysis. We have removed the reformulated metrics from our list of novel contributions in the paper. Accordingly, we do not intend to rigorously study the benefits of high-dimensional methods for the community at large.
>
> However, we would like to emphasize that these tests on ImageNet-scale models would not have been possible without the reformulations. The original metrics required calculating, storing, and manipulating ~300 GB feature covariance matrices which crashed a couple of the systems we tried using to run these tests. So, although we do not quantitatively demonstrate the advantage of the reformulations and do not include this as a contribution, the reformulations did contribute to the success of our analysis.
>
> *“I am not sure high-dimensionality is mandatory in representation similarity metric analysis because a feature dimension of a single layer of many Transformer-based models is from 768 to 2048…”*
>
> We are excited to extend this work to BERT and plan on including BERT in our analysis for the camera-ready submission. Because we have eliminated our reformulations from the list of important contributions in this paper, we do not thoroughly investigate the importance of high-dimensional tests here. However, we agree that this is important future work. Your feedback in this regard has helped us recognize and reframe the most important aspects of our work.
>
> [1] Ding, F., Denain, J. S., & Steinhardt, J. (2021). Grounding Representation Similarity Through Statistical Testing. Advances in Neural Information Processing Systems, 34, 1556-1568.
>
> [2] Meyes, R., de Puiseau, C. W., Posada-Moreno, A., & Meisen, T. (2020). Under the hood of neural networks: Characterizing learned representations by functional neuron populations and network ablations. arXiv preprint arXiv:2004.01254.
>
> [3] Zhou, B., Sun, Y., Bau, D., & Torralba, A. (2018). Revisiting the importance of individual units in cnns via ablation. arXiv preprint arXiv:1806.02891.
>
> [4] Donnelly, J., & Roegiest, A. (2019, April). On interpretability and feature representations: an analysis of the sentiment neuron. In European Conference on Information Retrieval (pp. 795-802). Springer, Cham.
>
> [5] Zhou, B., Sun, Y., Bau, D., & Torralba, A. (2018). Revisiting the importance of individual units in cnns via ablation. arXiv preprint arXiv:1806.02891.

---

> ### Author Response · Authors · 2022-11-19
> **Individual reviewer response to u5hR (Part 2)**
>
> *“It is not clear how to select 10 classes from 1k classes of ImageNet. Overall, the experimental setup is not clear for reproducibility and understanding even if they provided the source code…”*
>
> This feedback definitely helped us to clarify our experiment design. We have added important information regarding our experimental setup and design to the revised manuscript. In the case of ImageNet, we randomly chose our classes from the 1000 possible classes for both MobileNet (10 random classes) and AlexNet (50 random classes). In addition, we added a new Figure 1 to our manuscript which details the steps we took during our experiments. For our camera-ready submission, we plan on modifying our source code to include explicit instructions and documentation for reproducing our results.
>
> *“Following [Ding et al. 2021], experiments on multiple seeds are required. Although the authors argue they validate their method on popular CNN models, AlexNet and MobileNetV2 are not popular nowadays. Because [Ding et al. 2021] presents the results on BERT, the authors need to consider the similar experiments.”*
>
> We agree that this can be a crucial step to improve the generalizability of our findings. For the camera-ready version of the paper we plan on extending our tests to BERT. Specifically, we would conduct our tests on ten BERT models trained on random seeds released by [2] and used in [1].
>
> *“This paper is not easy to follow. Overall flow of this paper seems to follow [Ding et al. 2021]. But the paper is not typical organization. For example, Conclusion looks far from highlighting the core arguments of the paper. There are too many contribution summary. Indeed, some of them is hard to be considered as contributions (4th and 5th)”*
>
> We recognize the importance of this feedback. If you have any other specific suggestions regarding organization, we would be happy to make those changes. To reiterate, we have reformatted the paper to make the following organizational improvements:
> * Restructuring our abstract, introduction, and contributions to better reflect our intended contribution outlined in the response to Shared Concern #1.
> * Clarifying the differences between our contributions and the contributions of [1]
> * Reducing the number of our contributions to two to reflect the purpose of our paper
> * Reformatting the paper to improve its use of space, specifically to reduce the whitespace around figures.
> * Rewriting the Conclusion section to specifically address our contributions.
> * For the camera-ready version we plan on thoroughly documenting our code to facilitate the reproducibility of our experiments.
> * We have also included a new Figure 1 which helps to clarify the steps taken in our experimental design.
>
> [1] Ding, F., Denain, J. S., & Steinhardt, J. (2021). Grounding Representation Similarity Through Statistical Testing. Advances in Neural Information Processing Systems, 34, 1556-1568.
>
> [2] R. Zhong, D. Ghosh, D. Klein, and J. Steinhardt. Are larger pretrained language models uniformly better? comparing performance at the instance level. arXiv preprint arXiv:2105.06020, 2021.

---

### Official Review · Reviewer_ZbX5 · 2022-10-25

**Confidence:** 2
**Correctness:** 3
**Technical Novelty And Significance:** 2
**Empirical Novelty And Significance:** 2
**Recommendation:** 5

**Clarity, Quality, Novelty And Reproducibility:**

- Clarity, Quality： The organization can be further improved.
- Novelty: I think the novelty of this paper is a little limited.
- Reproducibility : The paper has provided the code for reproducibility.

**Strength And Weaknesses:**

**Strength:**

- The reformulation of CKA, Procrustes can be applied to high-dimentional representations.
- Some observations are new compared to existing studies.

**Weakness:**

- I think this paper is a simple extension of [1] which follows the same testing protocal. The observations are also empirical without deeper explanation.
- Apart from Image-net pretraining and classification, more experiments should be performed on different tasks to further verify the claims in this paper.
- The organization can be improved. For example, Page 8 has so much blank areas.

**Summary Of The Paper:**

This paper conducts a series of experiments to study how well each metric captures changes to representations. Linear decoding probes and network performance are choosen to be the measurements. Authors also propose several modification to existing metrics to make it possible to evaluate high-diemntional representations with low computational cost. Experiments are conducted on AlexNet and MobileNetV2 pre-trained with ImageNet.


**Summary Of The Review:**

- This paper has merits on reformulating existing metrics for advancing calculating high-dimentional representations and conducting experiments to observe the correlation between metric and functionality. Based on the experiments, authors find some new observations. However, I think the experiments are not sufficient to support the claims. The overall contribution is not enough for accepting this paper.

---

> ### Author Response · Authors · 2022-11-19
> **Individual reviewer response to ZbX5**
>
> Thank you for the positive feedback you presented regarding the usefulness of our reformulated metrics and the novelty of observations we make in our experiments. We would like to take the opportunity to address your constructive comments directly:
>
> *“I think this paper is a simple extension of [1] which follows the same testing protocol.”*
>
> We agree that the novelty of our work was not adequately presented in the original version of our manuscript. In our revised manuscript we have directly emphasized our novel contribution: a causal measure of how changes to a representation affect network performance and whether network performance changes are captured by a representational similarity metric. In short, commonly used representational similarity metrics do not capture changes in network performance very well.
>
> More specifically, we aim to test how sensitive representation similarity metrics are to network function. Studies that remove features from representations in trained networks have revealed a weak link between the relevance of a feature for decoding and its effect when removed from the network [2-5]. These studies test the causal relevance of information in representations for function. So, we wanted to conduct experiments to determine if representation similarity metrics were sensitive to this causal information. In our tests we directly compare non-causal linear probe accuracy changes on in-distribution inputs to causal network performance changes on the same inputs. On the other hand, [1] do not test this direct comparison. Instead, they perform more general stress tests on similarity metrics using either in-distribution linear probes or out-of-distribution network performance scores. Through this direct comparison, we are able to show that similarity metrics are significantly less sensitive to the causal features of representations that are actually used by the network compared to its decodable features.
>
> [1] Ding, F., Denain, J. S., & Steinhardt, J. (2021). Grounding Representation Similarity Through Statistical Testing. Advances in Neural Information Processing Systems, 34, 1556-1568.
>
> [2] Meyes, R., de Puiseau, C. W., Posada-Moreno, A., & Meisen, T. (2020). Under the hood of neural networks: Characterizing learned representations by functional neuron populations and network ablations. arXiv preprint arXiv:2004.01254.
>
> [3] Zhou, B., Sun, Y., Bau, D., & Torralba, A. (2018). Revisiting the importance of individual units in cnns via ablation. arXiv preprint arXiv:1806.02891.
>
> [4] Donnelly, J., & Roegiest, A. (2019, April). On interpretability and feature representations: an analysis of the sentiment neuron. In European Conference on Information Retrieval (pp. 795-802). Springer, Cham.
>
> [5] Zhou, B., Sun, Y., Bau, D., & Torralba, A. (2018). Revisiting the importance of individual units in cnns via ablation. arXiv preprint arXiv:1806.02891.

---

> ### Author Response · Authors · 2022-11-19
> **Individual reviewer response to ZbX5 (Part 2)**
>
> *“The observations are also empirical without deeper explanation.”*
>
> We agree that the observations we made in our first manuscript submission were mostly empirical. In our revised version, we have tried to emphasize the role that previous ablation studies play in motivating this work [5-8]. These studies motivate the need to test the sensitivity of similarity metrics to causal information present in representations and responsible for network function. Previous ablation studies provide a deeper explanation for motivating the current work, but what about deeper explanations of our results? Why do we see that existing representation similarity metrics are significantly less sensitive to causal network function? In the discussion we try to address this main result with a deeper explanation:
>
> Representation similarity metrics were designed to compare the linear geometric properties of two representation spaces. So, it is not surprising that similarity metrics correlate with changes in decoding accuracies. On the other hand, network performance measures of function reflect how the network utilizes representations. In this case, functionally similar representations are those representations that remain similar after a series of non-linear transformations through layers of the network. Perhaps it is not surprising that this non-linear notion of similarity is harder to capture using current similarity metrics. However, it is the ultimate goal of interpretability to link representation and non-linear network function.
>
> *“Apart from Image-net pretraining and classification, more experiments should be performed on different tasks to further verify the claims in this paper.”*
>
> We completely agree. In our work, we perform 16 ablations per layer for each of 10 randomly chosen ImageNet classes on two CNN models. The two CNN models differ in architecture, depth, and generalization performance. In our revised manuscript we have included updated results for 50 randomly chosen ImageNet classes in AlexNet. This addition extends the scope of the findings to a diverse and representative sample from ImageNet. However, we recognize the concern that the results should be extended to more models, tasks, and random seeds. For the camera-ready version of the paper we plan on extending our tests to BERT. Specifically, we would conduct our tests on the same ten BERT models released by [4] and used in [1].
>
> [1] Ding, F., Denain, J. S., & Steinhardt, J. (2021). Grounding Representation Similarity Through Statistical Testing. Advances in Neural Information Processing Systems, 34, 1556-1568.
>
> [2] Kriegeskorte, N., Mur, M., & Bandettini, P. A. (2008). Representational similarity analysis-connecting the branches of systems neuroscience. Frontiers in systems neuroscience, 4.
>
> [3] Hayne, L., Suresh, A., Jain, H., Kumar, R., & Carter, R. M. (2022). Much Easier Said Than Done: Falsifying the Causal Relevance of Linear Decoding Methods. arXiv preprint arXiv:2211.04367.
>
> [4] R. Zhong, D. Ghosh, D. Klein, and J. Steinhardt. Are larger pretrained language models uniformly better? comparing performance at the instance level. arXiv preprint arXiv:2105.06020, 2021.
>
> [5] Meyes, R., de Puiseau, C. W., Posada-Moreno, A., & Meisen, T. (2020). Under the hood of neural networks: Characterizing learned representations by functional neuron populations and network ablations. arXiv preprint arXiv:2004.01254.
>
> [6] Zhou, B., Sun, Y., Bau, D., & Torralba, A. (2018). Revisiting the importance of individual units in cnns via ablation. arXiv preprint arXiv:1806.02891.
>
> [7] Donnelly, J., & Roegiest, A. (2019, April). On interpretability and feature representations: an analysis of the sentiment neuron. In European Conference on Information Retrieval (pp. 795-802). Springer, Cham.
>
> [8] Zhou, B., Sun, Y., Bau, D., & Torralba, A. (2018). Revisiting the importance of individual units in cnns via ablation. arXiv preprint arXiv:1806.02891.

---

### Official Review · Reviewer_TGPW · 2022-11-02

**Confidence:** 2
**Correctness:** 4
**Technical Novelty And Significance:** 2
**Empirical Novelty And Significance:** 2
**Recommendation:** 5

**Clarity, Quality, Novelty And Reproducibility:**

Some questions on clarity:
The CNNs are trained on ImageNet. Is ImageNet also used as a test set for evaluating the metrics?
How do these functionality measures compare to the study performed by Ding et al. (2021)?



**Strength And Weaknesses:**

Strengths

The paper is well written - the theory and results are presented clearly. The paper presents interesting insights into the performance of various similarity metrics on representations generated by CNNs with two functionality measures.

Weaknesses

The novelty/originality is unclear.  The paper appears to extend the work of Ding et al. (2021) who performs similar comparisons of these metrics on ResNet with CIFAR-10 (omitting the analysis of these two functionality measures).  If the main contributions is the reformulation of the metrics and the inclusion of these two functionality measures - it would be useful to see how this compares to the work of Ding et al. (2021)  e.g. the computational resources used under the original formulations on ResNet or AlexNet vs the reformulation.

**Summary Of The Paper:**

This paper performs a comparative study of three representation similarity metrics (CKA, Procrustes and PWCCA) evaluated on representations generated by AlexNet and MobileV2Net. This work extends Ding et al. (2021) (which studies the same metrics on ResNet with CIFAR-10 test images and language models). The main contribution is the reformulation of these metrics to enable efficient computation on larger models with larger images. Further contributions include the comparison of the evaluation of these metrics on two functionality measures: linear probing accuracies and network performance deficits.

**Summary Of The Review:**

This work provides some interesting insights on representation similarity metrics evaluated on CNNs. The novelty/originality needs further clarification, especially with respect to Ding et al. (2021).

---

> ### Author Response · Authors · 2022-11-18
> **Individual response to reviewer TGPW**
>
> Thank you for the positive feedback you presented regarding the writing style and the interesting insights generated by our work. We would like to take this opportunity to respond directly to your constructive comments:
>
> *“The novelty/originality is unclear … ”*
>
> We agree that the novelty of our work was not adequately presented in our first manuscript. In our revised manuscript we have directly emphasized our novel contributions. We aim to portray this work as a unique set of experiments that does not extend the work of [1], but instead tests a fundamentally different hypothesis not addressed by the tests in [1]. Specifically, we aim to test how sensitive representation similarity metrics are to causal changes in network function. Studies that remove features from representations in trained networks have revealed a weak link between the relevance of a feature for decoding and its effect when removed from the network [2-5]. These studies test the causal relevance of information in representations for function. So, we wanted to conduct experiments to determine if representation similarity metrics were sensitive to this causal information. In our tests we directly compare non-causal linear probe accuracy changes on in-distribution inputs to causal network performance changes on the same inputs. On the other hand, [1] do not test this direct comparison. Instead, they perform more general stress tests on similarity metrics using either in-distribution linear probes or out-of-distribution network performance scores. Through this direct comparison, we are able to show that similarity metrics are significantly less sensitive to the causal features of representations that are actually used by the network compared to its decodable features.
>
> *“Is ImageNet also used as a test set for evaluating the metrics?”*
>
> Yes. The CNNs are trained on the training set of ImageNet and the metrics are evaluated using images from ImageNet’s validation set.
>
> *“How do these functionality measures compare to the study performed by Ding et al. (2021)?”*
>
> In our tests we directly compare non-causal linear probe accuracy changes on in-distribution inputs to causal network performance changes on the same inputs. On the other hand, [1] do not test this direct comparison. Instead, they perform more general stress tests on similarity metrics using either in-distribution linear probes or out-of-distribution network performance scores. Through this direct comparison, we are able to show that similarity metrics are significantly less sensitive to the causal features of representations that are actually used by the network compared to its decodable features.
>
> [1] Ding, F., Denain, J. S., & Steinhardt, J. (2021). Grounding Representation Similarity Through Statistical Testing. Advances in Neural Information Processing Systems, 34, 1556-1568.
>
> [2] Meyes, R., de Puiseau, C. W., Posada-Moreno, A., & Meisen, T. (2020). Under the hood of neural networks: Characterizing learned representations by functional neuron populations and network ablations. arXiv preprint arXiv:2004.01254.
>
> [3] Zhou, B., Sun, Y., Bau, D., & Torralba, A. (2018). Revisiting the importance of individual units in cnns via ablation. arXiv preprint arXiv:1806.02891.
>
> [4] Donnelly, J., & Roegiest, A. (2019, April). On interpretability and feature representations: an analysis of the sentiment neuron. In European Conference on Information Retrieval (pp. 795-802). Springer, Cham.
>
> [5] Zhou, B., Sun, Y., Bau, D., & Torralba, A. (2018). Revisiting the importance of individual units in cnns via ablation. arXiv preprint arXiv:1806.02891.

---

### Author Response · Authors · 2022-11-18
**Thank you to all reviewers. General response.**

Thank you to all the reviewers for your thoughtful and constructive feedback. We are glad to see that all three reviewers highlighted strengths in our paper. The reviewers emphasized that the paper was “well written” and “presented clearly” (TGPW), that the reformulated metrics “can be applied to high-dimensional representations” (ZbX5) and is “a significant topic” of study (u5hR), and that the analysis is “valuable” (u5hR), presents “new observations” (ZbX5) and “provides some interesting insights” (TGPW).

We also recognize that the reviews contain significant critiques. We recognize the value of those critiques and believe we are in a position to make significant progress to alleviating those concerns. Primarily, reviewers were concerned that the novelty of this project was not clear. Our lack clarity stemmed from our desire to emphasize overcoming a technical hurdle. The novelty of our paper stems from the use of a causal test of representation utility to compare representational similarity metrics. In the latest version of the manuscript, we highlight that our tests address a fundamentally different question not investigated by previous works (including [1]). Specifically, we test how sensitive representation similarity metrics are to representation features responsible for network function.

Studies that remove features from representations in trained networks have revealed a weak link between the relevance of a feature for decoding and its effect when removed from the network [2-5]. These studies test the causal relevance of information in representations for function. So, we wanted to conduct experiments to determine if representation similarity metrics were sensitive to this causal information. In our tests we directly measure linear probe accuracy changes on in-distribution inputs to network performance changes on the same inputs. On the other hand [1] do not test this direct comparison. Instead, they perform more general stress tests on similarity metrics using either in-distribution linear probes or out-of-distribution network performance scores. Through this direct comparison, we are able to show that similarity metrics are significantly less sensitive to the functional features actually used by the network compared to the decodable features of a representation. It is a significant step closer to true causal tests of the relationship between representation and function. We first provide initial answers to the reviewers individual concerns below and include the rewritten manuscript.

[1] Ding, F., Denain, J. S., & Steinhardt, J. (2021). Grounding Representation Similarity Through Statistical Testing. Advances in Neural Information Processing Systems, 34, 1556-1568.

[2] Meyes, R., de Puiseau, C. W., Posada-Moreno, A., & Meisen, T. (2020). Under the hood of neural networks: Characterizing learned representations by functional neuron populations and network ablations. arXiv preprint arXiv:2004.01254.

[3] Zhou, B., Sun, Y., Bau, D., & Torralba, A. (2018). Revisiting the importance of individual units in cnns via ablation. arXiv preprint arXiv:1806.02891.

[4] Donnelly, J., & Roegiest, A. (2019, April). On interpretability and feature representations: an analysis of the sentiment neuron. In European Conference on Information Retrieval (pp. 795-802). Springer, Cham.

[5] Zhou, B., Sun, Y., Bau, D., & Torralba, A. (2018). Revisiting the importance of individual units in cnns via ablation. arXiv preprint arXiv:1806.02891.

---

### Author Response · Authors · 2022-11-18
**Addressing shared reviewer concerns**

First, we would like to address some of the shared concerns of the reviewers:

1. Shared concern by TGPW, ZbX5, and u5hR: “This paper is a simple extension of [1]”

We can very much appreciate this concern. We designed our experiments using the same testing framework as [1], but not to extend that work. Instead, we used different measures of function to test how sensitive representation similarity metrics are to network function. To test this, it was necessary to measure in-distribution linear probe and network performance changes on the same input examples. This direct comparison would not have been possible using the measures collected in [1]. Additionally, we use a different method of perturbing representations from [1]. Namely, we systematically remove features from representations (ablation) to perturb them.

The revised manuscript includes the distinctions between our work and previous work and highlights our unique contributions to the study of representation similarity metrics. The introduction section has been revised to justify the need for our chosen functional measures based on the problem we aim to solve. It also includes a justification for using ablation. The problem we aim to address is supported by previous work on ablation and representation similarity. We have reframed the introduction to emphasize this problem and our unique approach for addressing it. The discussion section has been revised to detail the difference between our work and [1]. While we adopt their framework for testing representation similarity metrics, we address a fundamentally different problem using different perturbation and functional measures. Our experiments reveal a new finding: that similarity metrics are significantly less sensitive to causal differences in network performance induced by ablation than decodable changes induced by ablation. This finding emphasizes an important set of tests necessary for improving similarity metrics for the purpose of interpretability.

2. Shared concern by TGPW and u5hR: “Experiments on higher dimensional feature similarity might be required”

Because of the novelty of the reformulated metrics necessary for performing our experiments on larger images, we overemphasized the reformulations. As described above, our experiments were structured to test whether existing similarity metrics actually capture functional changes induced by ablation. As outlined in the response to Shared Concern #1, we have emphasized in the revised manuscript the insensitivity of metrics to function and its implications for interpretability as our major contribution. As a result of this restructuring, we instead frame the reformulated metrics as a tool for facilitating our analysis. Accordingly, we do not intend to rigorously study the benefits of high-dimensional methods for the community at large.

3. Shared concern by ZbX5 and u5hR: “The organization can be improved”

We recognize the importance of this feedback. We have reformatted the paper to make the following organizational improvements:
* Restructuring our abstract, introduction, and contributions to better reflect our intended contribution outlined in the response to Shared Concern #1.
* Clarifying the differences between our contributions and the contributions of [1]
* Reducing the number of our contributions to two to reflect the purpose of our paper
* Reformatting the paper to improve its use of space, specifically to reduce the whitespace around figures.
* Rewriting the Conclusion section to specifically address our contributions.
* For the camera-ready version we plan on thoroughly documenting our code to facilitate the reproducibility of our experiments.
* We have also included a new Figure 1 which helps to clarify the steps taken in our experimental design.

4. Shared concern by ZbX5 and u5hR: “More experiments should be performed on different tasks to further verify the claims in this paper”

In our work, we perform 16 ablations per layer for each of 10 randomly chosen ImageNet classes on two CNN models. The two CNN models differ in architecture, depth, and generalization performance. In our revised manuscript we have included updated results for 50 randomly chosen ImageNet classes in AlexNet. This addition extends the scope of the findings to a diverse and representative sample from ImageNet. However, we recognize the reviewers’ concerns that the results should be extended to more models, tasks, and random seeds. For the camera-ready version of the paper we plan on extending our tests to BERT. Specifically, we would conduct our tests on the same ten BERT models released by [2] and used in [1].

[1] Ding, F., Denain, J. S., & Steinhardt, J. (2021). Grounding Representation Similarity Through Statistical Testing.

[2] R. Zhong, D. Ghosh, D. Klein, and J. Steinhardt. Are larger pretrained language models uniformly better? comparing performance at the instance level.

---

### Decision · Program_Chairs · 2023-01-20

**Decision:**

Reject

**Justification For Why Not Higher Score:**

Lack of novelty

**Justification For Why Not Lower Score:**

n/a

**Metareview: Summary, Strengths And Weaknesses:**

Although the reviewers noted that the paper is well written and the topic is relevant, I do not recommend accepting the paper in its current form. The main issue, raised by all reviewers, is that the work is quite similar to Ding et al. Some of the reviewers would also have liked to see experiments with higher dimensional feature similarity, and experiments on more datasets.

**Summary Of Ac-Reviewer Meeting:**

n/a